# The Reliability and Validation of the Children’s Eating Attitude Test among Chinese Samples

**DOI:** 10.3390/ijerph20010738

**Published:** 2022-12-31

**Authors:** Ying Huang, Chang Wang, Lian Tong

**Affiliations:** School of Public Health, Key Laboratory Public Health Safety of Chinese Ministry of Education, Fudan University, Shanghai 200437, China

**Keywords:** children’s eating attitudes test, Chinese children, validation, BMI

## Abstract

The purpose of this study was to examine the psychometric properties of the Children’s Eating Attitude Test (ChEAT) in a Chinese sample. A total of 906 children (mean age = 10.55, SD = 1.08) from three primary schools were evaluated by the ChEAT. Factor analysis was performed to examine the factor structure of the ChEAT. The children’s body mass index (BMI) was applied to assess the concurrent validity of the ChEAT. The ChEAT showed good internal consistency (Cronbach’s α = 0.819) and split-half reliability (0.816) in Chinese children. Exploratory factor analysis suggested a four-factor structure, consistent with previous studies, which explained 41.16% of the total variance. Confirmatory factor analysis indicated good construct validity for the Chinese sample. The Chinese version of the ChEAT showed evidence for reliability and validity to evaluate the eating attitudes and behaviors for Chinese children. The mean score of each factor of the ChEAT differed significantly among different genders and BMI groups. Overweight girls had more eating disorder problems than normal-weight girls, and boys with lower BMI showed higher social eating pressure.

## 1. Introduction

An eating disorder is a severe chronic mental illness defined by abnormal eating habits that negatively affect a person’s physical or mental health [1], associated with other health issues, such as overweight, obese, depression, and substance abuse [2]. According to the Diagnostic and Statistical Manual of Mental Disorders, Fifth Edition (DSM-5), eating disorders comprise a range of psychological conditions including anorexia nervosa, bulimia nervosa, binge eating disorder, and other specified feeding and eating disorders [3]. Eating disorders are most common during adolescence and young adulthood [4,5], which have been reported to start as early as the age of 9 years [6]. Eating disorders are often reported in school-aged children worldwide. One Austrian survey among adolescents aged 10–18 years reported that 30.9% of girls and 14.6% of boys were screened to be at risk for eating disorders [5]. A similar prevalence was found in European samples, including German children [7], Spanish adolescents [8], and Finnish adolescents [9]. In east Asia, the prevalence of eating disorders was found to be 9.1% among Korean children aged 7–13 years old [10]. Several studies indicated that the prevalence of eating disorders varied from 1.3% to 5.2% in Chinese adolescents and young adults aged 15–24 years old, and 6.3% in girls aged 12–17 years old [11,12,13].

Abnormal eating attitudes and behaviors in childhood may eventually evolve into eating disorders, which are likely to continue into young adulthood [4,14,15]. Early detection of abnormal eating attitudes and behaviors is a crucial step for preventing the occurrence of eating disorders. In China, there is a lack of reliable and validated tools for screening eating disorders in children currently. Most studies on eating disorders conducted among Chinese adolescents and adults have used the Eating Attitudes Test (EAT) and Eating Disorder Inventory (EDI) as investigation tools [16,17,18]. For screening eating disorders in children, the Child Eating Behavior Questionnaire (CEBQ) is a parent-report questionnaire designed in the U.K. It included responsiveness to food, enjoyment of food, satiety responsiveness, slowness in eating, fussiness, emotional overeating, and emotional undereating [19], but few studies have been conducted on Chinese children. The lack of an efficient screening tool may be an obstacle to the early detection of eating disorder symptoms in Chinese children. 

A reliable scale that can identify and assess the severity of these problems is essential for studying eating behaviors. The EAT is a self-report screening questionnaire designed to detect eating disorder psychopathology in adults [20,21]. The EAT has been found to reliably identify potential cases of eating disorders in non-clinical populations [22]. The adult version of the measure has been used with older adolescents but is not suitable for use with younger children [23].

The Children’s Eating Attitude Test (ChEAT) is a simplified version of the EAT designed to specifically measure a wide range of problematic eating attitudes and behaviors among children and adolescents aged 8–13 years [23]. The basic psychometric properties of the ChEAT are similar to those of the adult EAT, making it a valuable measure for capturing children’s eating attitudes and behaviors in different cultures [24]. Many studies have shown that the ChEAT has adequate test–retest reliability and internal reliability in Western cultures [25,26,27]. For instance, American research found that the scale had adequate internal reliability with a Cronbach’s α of 0.87 [25]. Few studies have been conducted on Asian children, with the exception of one study among Japanese children aged 10–15 years, which displayed good internal consistency [28]. 

Although the ChEAT has displayed good psychometric properties in other ethnic groups, ongoing work is required to confirm its reliability and validity for Chinese children, considering the food culture differences between Western and Eastern countries. Therefore, the present study sought to investigate the psychometric properties of the ChEAT in a sample of Chinese children. Our findings will facilitate cross-cultural comparative studies on eating disorders in children.

## 2. Materials and Methods

### 2.1. Participants

A total of 906 children (53.42% boys and 46.58% girls) from grades three to five and their parents were recruited by stratified clustered sampling from three primary schools located in Shanghai, China. The average age of the children was 10.55 (SD = 1.08) years. The schools selected by stratified random sampling were located in central urban or suburban areas of Shanghai and represented different socioeconomic levels. The parents of School A had the highest socioeconomic status (SES; e.g., high educational attainment, high household income, and age), and the parents of School C had the lowest SES. Informed consent was obtained from primary schools and legal parents of all participating students. 

### 2.2. Measures

Both a student self-reported questionnaire and a parent-completed questionnaire were used in this study. The two questionnaires were matched by the student identification number given by the school. The ChEAT was obtained with a student self-reported questionnaire. Other demographic information of the child was collected by a parent-completed questionnaire. The specific assessment instruments used in this study were as follows.

#### 2.2.1. ChEAT

The ChEAT consists of 26 items to describe behavior and emotional states connected with eating and body image. Each item is rated on a six-point Likert scale including “never,” “rarely,” “sometimes,” “often,” “usually,” and “always.” The most symptomatic response is recorded as a score of 3 (“always”), the next as 2 (“usually”), and the next as 1 (“often”), and the remaining options (“sometimes,” “rarely,” and “never”) receive a score of 0. Therefore, the total score of the ChEAT ranges from 0 to 78, while a higher score indicates more eating behavior disturbance. The cut-off point is 20 to distinguish normal and abnormal groups of eating behaviors after verification [21]. The original version was translated into Chinese by professors. Back translation was performed by a bilingual psychiatric graduate from the USA, and overlap with the English original version was found.

#### 2.2.2. Children’s BMI

The body mass index (BMI) was calculated as weight divided by height squared (kg/m^2^) measured with the participant wearing indoor clothing and standing barefoot. The height and weight were measured by a healthcare provider in the school hospital. The International Obesity Task Force (IOTF) body mass index cut-off points for Asian children were used in this study [29]. The definition provides cut-off points at 1-month intervals from age 2 to 18 years, for boys and girls separately. According to the cut-off points for the age- and gender-specific BMI of children, children were divided into six groups: Thinness grade 3, Thinness grade 2, Thinness grade 1, Fit, Overweight, and Obese.

### 2.3. Statistical Analysis

The data were analyzed using the Statistic Analysis System (SAS) 9.3 (SAS Institute Inc., Cary, NC, USA), Stata 16.0 (StataCorp LLC, College Station, TX, USA), and Mplus version 7 (Muthén and Muthén, Los Angeles, CA, USA).

The score for the ChEAT total scale and each subscale had a non-normal distribution, so the median and interquartile range (IQR) were selected for comparing gender differences. Both the median and mean scores of the ChEAT were reported to show the whole picture of samples. Mood’s median test was used to compare the medians of different gender groups. The data were assigned to two groups, and chi-square tests were conducted, one consisting of data whose values were higher than the median value and the other consisting of data whose values were at the median or below [30,31]. The *t* test was used to determine the equality of means in different gender groups [32].

Exploratory factor analysis was used to determine the number of factors for the ChEAT-26 in the sample of Chinese children. A model’s goodness of fit was assessed by the Kaiser–Meyer–Olkin (KMO) and Bartlett’s sphericity tests, considering values over 0.70 and *p* < 0.05 as acceptable. The construct validity of the ChEAT-26 was tested using confirmatory factor analysis (CFA) to ensure the theoretical structure of the scale [33]. Fit indices were examined to evaluate the model fit, including a chi-square test of model fit, root-mean-square error of approximation (RMSEA), comparative fit index (CFI), and Tucker–Lewis index (TLI) [34]. A CFA with the ChEAT was performed using the weighted least squares mean and variance adjusted (WLSMV) robust estimation method. The WLSMV is suitable for non-normally distributed variables and categorical or ordered data [33]. Internal reliability, split-half reliability, and test–retest reliability were used to measure the reliability of the ChEAT. Correlation analysis of each of the scales and the BMI of the children in the study was conducted. As the ChEAT score of the different BMI groups did not conform to the normal distribution, the Kruskal–Wallis test was used to compare groups by total and subtotal ChEAT-26 scores.

## 3. Results

### 3.1. Demographic Information of Children

This study used a mixed sample of different socioeconomic strata from China. Table 1 shows the gender, grade, school, age, parent’s education level, and household income information of the participants. The cut-off point of 20 was used to distinguish normal and abnormal groups of eating behaviors after verification [21]. According to this cut-off point, 10.71% of children had positive eating disorder symptoms in the present study. No significant differences were found between the two groups regarding gender, grade, school, or age. A χ-quadrate analysis also showed no significant differences in family status between the two groups.

### 3.2. Exploratory Factor Analysis

The 26-item ChEAT was subjected to exploratory factor analysis. The Kaiser–Meyer–Oklin test presented a value of 0.880. Bartlett’s test of sphericity indicated statistical significance (*p* < 0.001), which meant that the factorability of the correlation matrix was supported. The internal consistency coefficient (Cronbach’s α) was 0.819 for the sample of Chinese children.

Exploratory factor analysis indicated the presence of five components with eigenvalues exceeding 1, explaining 53.95% of the total variance, and a screen plot was generated (Figure 1). In consideration of the items’ interpretability and Cattell’s scree plot, we selected four factor structures that explained 41.16% of the variance (Table 2). The labels for the factor were as follows: Factor 1 was named “fear of getting fat”; Factor 2 was named “dieting and purging behaviors”; Factor 3 was named “bulimia and food preoccupation”; Factor 4 was named “social pressure to eat”.

### 3.3. Confirmatory Factor Analysis (CFA)

The ChEAT with the four-factor structure was analyzed by CFA. Considering the modification indexes, paths were added between some items that are correlated and theoretically explicable in the model. After adjusting the model, the goodness of fit for the ChEAT was acceptable (χ^2^/df = 2.943, RMSEA  =  0.046, CFI  =  0.883, and TLI  =  0.869), indicating good construct validity for the Chinese sample (Figure 2 and Table A1).

### 3.4. Reliability of the ChEAT

The Cronbach’s α was 0.819 for the ChEAT, indicating good internal consistency. Cronbach’s α for the four subscales of children ranged from 0.536 (social pressure to eat) to 0.728 (fear of getting fat). The split-half reliability coefficient was 0.816 for the ChEAT (Table 3).

### 3.5. Descriptive Statistics of the ChEAT

In Factor 1, girls (median = 3, IQR: 0–7) showed more fear of getting fat than boys (median = 2, IQR: 0–6) (χ^2^ = 10.630, *p* < 0.01). In Factor 3 of bulimia and food occupation, the median test showed that the difference was statistically significant between girls and boys (χ^2^ = 6.1325, *p* < 0.05). In Factor 4 of social pressure to eat, girls (median = 2, IQR: 0–4) also expressed a significantly higher difference than boys (median = 1, IQR: 0–3, χ^2^ = 3.850, *p* < 0.05). The median score of the ChEAT in the whole sample was 7 (IQR: 3–13) without a significant difference in the two gender groups. There were no significant gender differences in Factor 2 of dieting and purging behaviors and total score. Except for Factors 3 and 4, the results of the mean *t* test were consistent with the median test. The mean scores of the ChEAT are shown in Table 4.

### 3.6. The Associations between ChEAT Scores and BMI

The mean BMI was 17.69 for the total sample (SD = 3.47), and it was significantly lower in girls than in boys (M = 17.22 ± 3.30 vs. M = 18.18 ± 3.54, *p* < 0.05). The correlation analysis suggested that the total score of the ChEAT was positively correlated with BMI (r = 0.114, *p* = 0.002). Children with a higher score on Factor 1 (fear of getting fat; r = 0.268, *p* < 0.001) as well as Factor 2 (dieting and purging behaviors; r = 0.140, *p* < 0.001) had a higher BMI. Girls showed similar correlations between the ChEAT and BMI, while for the boy’s sub-sample, those with a higher score on Factor 2 of dieting and purging behaviors had a lower BMI (r = −0.105, *p* = 0.043; Table 5). On the contrary, children with a higher score on Factor 4 (social pressure to eat; r = 0.161, *p* = 0.001) presented a lower BMI. The correlation analysis suggested no significant differences in Factor 3 (bulimia and food occupation) among all children.

### 3.7. The ChEAT Scores in Different BMI Groups

The differences in eating behavior dimensions between distinct BMI groups were analyzed. It was found that the scores on Factor 1 and Factor 2 increased significantly with the BMI group, from thinness grade 3 to obesity. Factor 4 showed the inverse relationship. In the girl’s sub-sample, significant mean score differences were found in Factor 1 (χ^2^ = 42.595, *p* < 0.0001), Factor 2 (χ^2^ = 12.375, *p* = 0.030), and the total score (χ^2^ = 13.357, *p* = 0.020), increasing distinctly through the six BMI groups. In the boy’s sub-sample, significant differences were found in Factor 1 (χ^2^ = 20.511, *p* = 0.001), Factor 2 (χ^2^ = 11.672, *p* = 0.040), and Factor 4 (χ^2^ = 22.319, *p* = 0.001), and the BMI increased significantly with ascending Factor 1 scores and descending Factor 4 scores (Table 6).

## 4. Discussion

This study presented the psychometric properties and the validation of the ChEAT questionnaire in a large representative sample of young children in China. We examined the internal reliability, construct validity, and concurrent validity. The Chinese version of the ChEAT showed high internal consistency, reliability, and validity. The internal reliability (α = 0.819) was slightly higher than those reported in previous studies conducted in other cultures [28,35]. The split-half reliability coefficient was good, reaching 0.816 for the entire scale. This Chinese ChEAT comprises four factors accounting for 41.16% of the total variance. The number of factors was consistent with the Finnish version and the Polish version [27,36]. The composition of the factors also indicated a high level of consistency with the Japanese version, though the Japanese version comprised five factors. The items for Factor 2 of dieting and purging behaviors in the Chinese version were especially similar to the items for the factors of dieting and purging in the Japanese version. Factor 3 of bulimia and food preoccupation had the same composition in the two versions except item 24, “I like my stomach to be empty.” The composition of Factor 4, “social pressure to eat,” in the Chinese version was also highly similar to the Japanese version. The consistency of composition may be due to the similarity of the food culture in East Asia.

In this study, the convergent validity and discriminant validity of the ChEAT were confirmed. Considering the modification indexes, the error of item 11 (I think a lot about wanting to be thinner) was correlated with the error of item 1 (I am scared about being overweight); the error of item 19 (I can show self-control around food) was correlated with the error of item 18 (I think that food controls my life). The above findings are likely due to some overlap of the two items.

According to the cut-off point of 20, 10.71% of children were identified as the eating disorder symptoms group in the present study, which is slightly lower than the previous study (13.49%) [26]. The mean score of the ChEAT (9.21 ± 9.11) is slightly higher than the means reported in the Portuguese version (8.61 ± 7.03) and Finnish version (2.55 ± 4.22). According to previous research, the mean scores of the ChEAT have been observed to be higher in large cities than in small- and medium-sized cities [28,36]. Our participants came from a large city, which could partly explain our mean score being higher than those in previous studies. Urban children are exposed more extensively to the media, including social media, and influenced by the “thin body ideal” from Western culture than rural children. The media is a primary source of pressure relating to body dissatisfaction and may even predict body change behaviors [37]. A previous study found a strong and consistent association between social media use and eating concerns [38]. According to a 25-country European Union Kids Online survey, 10% of children aged 9–16 years had seen eating disorder sites online, with girls being more commonly exposed to such material than boys [39,40].

In this research, the ChEAT scores of boys and girls showed differences in many respects; for example, girls showed more fear of getting fat and social pressure to eat than boys. In Factor 3 of bulimia and food preoccupation, boys scored higher than girls. The total score of the ChEAT and Factor 3 did not differ significantly between the two gender groups. A previous study found that among fourth- and sixth-graders, girls indicated more concern than boys about being or becoming overweight [41]. Furthermore, girls showed significantly higher body dissatisfaction than boys at 7, 9, and 12 years of age [42]. These findings indicated that the Chinese version of ChEAT-26 has a gender sensitivity.

It is well known that people with eating disorders usually have problems with body weight, being either overweight or underweight. BMI is independently associated with disturbed eating attitudes and behaviors, as it has previously been found that a higher BMI is a risk factor for eating disorder symptoms [43,44]. Whether among girls or boys, there are strong direct associations between BMI and both dieting and disordered eating behaviors, and these associations remain significant after adjusting for sociodemographic characteristics [45]. Therefore, the correlations between the ChEAT and the BMI were used to assess concurrent validity in this study. It was found that the mean scores of the ChEAT for Factor 1, “fear of getting fat,” and Factor 2, “dieting and purging behaviors,” showed significant differences among the six BMI groups. Body concerns are psychological variables and are partly based on actual body mass [46]. Previous studies reported that overweight children are more likely to have disordered eating symptoms [47,48]. Our finding of overweight girls having higher ChEAT scores than normal-weight girls is in line with previous studies [48]. Research with adults and adolescents suggests that body dissatisfaction may fully mediate the inconsistently observed relationships between eating disorder symptoms and BMI [42,49,50].

In addition, the present study found that boys with lower BMI showed higher social eating pressure, which is similar to the results of the Finnish study [27]. However, there was no significant correlation between social eating pressure and BMI among girls. The Chinese son preference, which means that boys have more value than girls in China, might help explain this phenomenon. It is believed that boys should be superior to girls in terms of physical strength and reach Chinese conceptions of masculine ideals [51]. Thus, boys with low BMI face more social eating pressure than girls. The higher prevalence of obesity and overweight in boys than girls may be a result of this pressure [52,53]. Another possible explanation is that boys interpret the questions of the ChEAT differently than girls. For example, item 13, “Others think I am too thin,” may be interpreted differently by boys than by girls [10].

Regardless of gender, there was no significant correlation between Factor 3, “bulimia and food preoccupation,” and BMI. This might be due to the differences in participant age, as the Spanish version selected schoolchildren aged 13–17 years [54] and the Japanese version selected children aged 10–15 years [28], whereas we selected children whose mean age was 10.55. In the Polish version, the average age of children was 11.8 (standard deviation = 0.9), and there was a significant positive correlation between BMI and the “compulsive-bulimic” trait. Younger children may have had less of a chance to feel guilty about overeating, and it may take some time for binge eating to change the BMI. These findings suggest that the participants’ age should be considered in ChEAT26 factor analyses [28].

In summary, we assessed the consistency and reliability of the ChEAT among Chinese children. The Chinese version of the ChEAT is an applicable and valid scale for measuring eating attitudes and behaviors in Chinese children. Furthermore, as the only existing study on the ChEAT in China, our study provides a reference for deeper research on this subject.

The limitations of this study include that the children were limited to grades 3–5, which may affect the representativeness of samples. In addition, the proportion of eating problems in the population was not high. The proportion of 906 students who actually had eating problems may be low. However, as a validation study, this study was performed with an adequate sample size, which was greater than a ratio of 10 participants per item [55]. Our Cronbach’s alpha values for Factors 2–4 were much lower than 0.7 (0.536–0.699). A similar study conducted in Japan [28] showed similar results: Cronbach’s alphas for Factors 2–5 were between 0.39 and 0.62. The sample in this study comprised students with typical development, so the reliability of these factors was relatively low. For concurrent validity, correlation analysis between the ChEAT and the BMI was not comprehensive for the validity test, so future studies are encouraged to evaluate the concurrent validity of the Chinese version of the ChEAT by using multiple tools assessing eating disorders in children. We did not perform a structured diagnostic for participants who scored above the cut-off to distinguish types of eating disorder, such as anorexia nervosa, bulimia nervosa, and binge eating disorders, which does not permit us to draw further conclusions about the scale’s validity in samples with a specific eating disorder. However, the ChEAT can be used to screen school children for eating problems.

## 5. Conclusions

The Chinese version of the ChEAT showed evidence for reliability and validity to evaluate the eating attitudes and behaviors for Chinese children. The Chinese version of the ChEAT is a reliable and valid psychometric tool that may be useful in the assessment of children aged 9 to 13 years with abnormal eating attitudes. Our findings will facilitate cross-cultural comparative studies on eating disorders in children. The eating behavior differed significantly among different genders and BMI groups. Overweight girls had more eating disorder problems than normal-weight girls, and boys with lower BMI showed higher social eating pressure.

## Figures and Tables

**Figure 1 ijerph-20-00738-f001:**
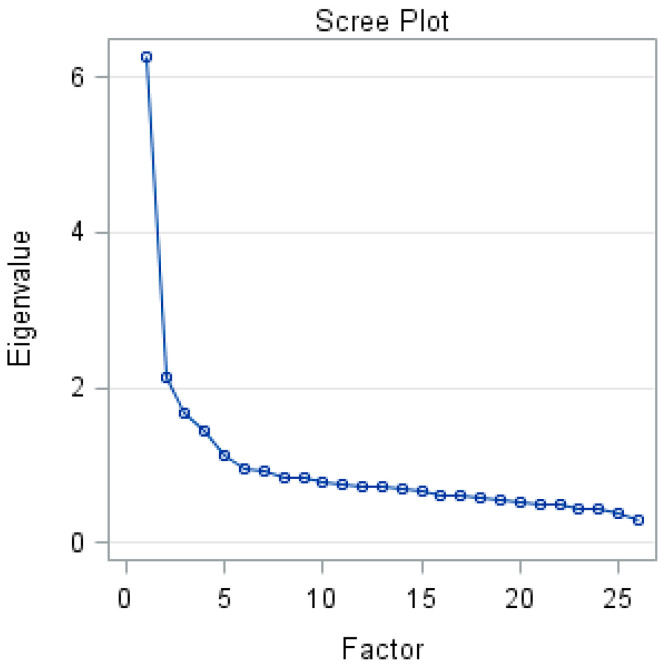
The Scree Plot of the exploratory factor analysis of ChEAT.

**Figure 2 ijerph-20-00738-f002:**
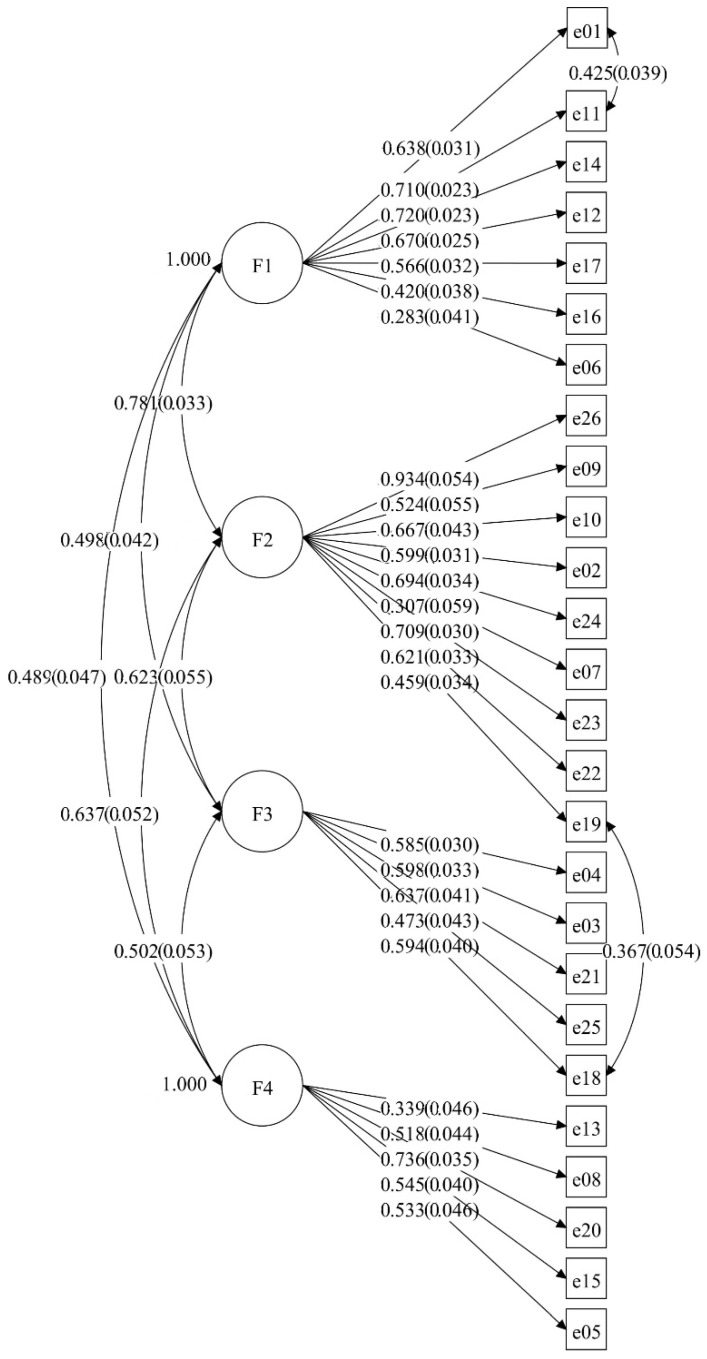
CFA results of the ChEAT.

**Table 1 ijerph-20-00738-t001:** Demographic information of participants distinguished by ChEAT.

Variable	Total, *n* (%)	Normal, *n* (%)	Abnormal, *n* (%)	χ^2^ **
Gender				
Boys	484 (53.42)	432 (47.68)	52 (5.74)	0.002
Girls	422 (46.58)	377 (41.61)	45 (4.97)	
Grade				
Three	323 (35.65)	287 (31.68)	36 (3.97)	2.80
Four	276 (30.46)	241 (26.60)	35 (3.86)	
Five	307 (33.89)	281 (31.02)	26 (2.87)	
School				
A	366 (40.40)	320 (35.32)	46 (5.08)	2.36
B	325 (35.87)	293 (32.34)	32 (3.53)	
C	215 (23.73)	196 (21.63)	19 (2.10)	
Age (NA * = 78, 8.61%)				
8–9	134 (16.18)	118 (14.25)	16 (1.93)	2.66
10	277 (33.45)	247 (29.83)	30 (3.62)	
11	245 (29.59)	216 (26.09)	29 (3.50)	
12	150 (18.12)	138 (16.67)	12 (1.45)	
13	17 (2.29)	17 (2.05)	2 (0.24)	
14–16	3 (0.36)	3 (0.36)	0 (0)	
Annual household income (USD; NA = 128, 14.13%)				
<3000	57 (7.33)	49 (6.30)	8 (1.03)	4.32
3000–6000	98 (12.60)	82 (10.54)	16 (2.06)	
6000–9000	151 (19.41)	135 (17.35)	16 (2.06)	
9000–12,000	111 (14.27)	100 (12.85)	11 (1.41)	
12,000–15,000	142 (18.25)	127 (16.32)	15 (1.93)	
≥15,000	219 (28.15)	199 (25.58)	20 (2.57)	
Mother’s education (NA = 93, 10.26%)				
Illiteracy/Primary school	96 (11.81)	86 (10.58)	10 (1.23)	0.70
Middle school	298 (36.65)	265 (32.60)	33 (4.06)	
High school	200 (24.60)	179 (22.02)	21 (2.58)	
University	214 (26.32)	192 (23.62)	22 (2.71)	
Post college	5 (0.62)	5 (0.62)	0 (0)	
Father’s education (NA = 89, 9.82%)				
Illiteracy/Primary school	31 (3.79)	27 (3.30)	4 (0.49)	2.48
Middle school	312 (38.19)	278 (34.03)	34 (416)	
High school	222 (27.17)	199 (24.36)	23 (2.82)	
University	233 (28.52)	208 (25.46)	25 (3.06)	
Post college	19 (2.33)	19 (2.33)	0 (0)	
Siblings (NA = 73, 8.06%)				
Single child	477 (57.26)	426 (54.76)	51 (6.56)	1.76
Non-single child	356 (42.74)	317 (40.75)	39 (5.01)	
Total	906 (100)	809 (89.30)	97 (10.71)	

Note: * NA means the number of missing data. ** None of *p* value chi-square tests meet the significant level of 0.05.

**Table 2 ijerph-20-00738-t002:** Exploratory Factor analysis for ChEAT in Chinese children.

No.	Items	Factor 1 Fear of Getting Fat	Factor 2 Dieting and Purging Behaviors	Factor 3 Bulimia and Food Occupation	Factor 4 Social Pressure to Eat
11	I think a lot about wanting to be thinner	0.798			
01	I am scared about being overweight	0.728			
14	I think a lot about having fat on my body	0.669			
12	I think about burning up energy (calories) when I exercise	0.617			
17	I eat diet foods	0.496			
16	I stay away from foods with sugar in them	0.364			
06	I am aware of the energy (calorie) content in foods that I eat	0.317			
26	I have the urge to vomit after eating		0.740		
09	I vomit after I have eaten		0.734		
10	I feel very guilty after eating		0.577		
02	I stay away from eating when I am hungry		0.545		
24	I like my stomach to be empty		0.513		
07	I try to stay away from foods such as breads, potatoes, and rice		0.461		
23	I have been dieting		0.430		
22	I feel uncomfortable after eating sweets		0.418		
19	I can show self-control around food		0.355		
04	I have gone on eating binges where I feel that I might not be able to stop			0.709	
03	I think about food a lot of the time			0.658	
21	I give too much time and thought to food			0.596	
25	I enjoy trying new rich foods			0.588	
18	I think that food controls my life			0.585	
13	Other people think I am too thin				0.699
8	I feel that others would like me to eat more				0.671
20	I feel that others pressure me to eat				0.576
15	I take longer than others to eat my meals				0.402
5	I cut my food into small pieces				0.367

**Table 3 ijerph-20-00738-t003:** Reliability of the ChEAT.

Dimension	Item Number	Cronbach’s *α*	Split-Half Reliability
F1 Fear of getting fat	7	0.728	0.502
F2 Dieting and purging behaviors	9	0.699	0.700
F3 Bulimia and food occupation	5	0.645	0.617
F4 Social pressure to eat	5	0.536	0.442
Total	26	0.819	0.816

**Table 4 ijerph-20-00738-t004:** Descriptive statistics of ChEAT scores by children’s gender.

Gender	Median	IQR	*χ* ^2^	Mean ± SD	*t*
Total					
Boys	6	2–12	1.820	8.79 ± 9.08	−1.511
Girls	8	3–14		9.70 ± 9.13	
Total	7	3–13		9.21 ± 9.11	
Factor 1, fear of getting fat					
Boys	2	0–6	10.630 **	3.78 ± 4.42	−2.923 **
Girls	3	0–7		4.67 ± 4.68	
Total	3	0–7		4.20 ± 4.56	
Factor 2, dieting and purging behaviors					
Boys	0	0–2	0.499	1.49 ± 2.88	0.502
Girls	0	0–2		1.39 ± 2.87	
Total	0	0–2		1.44 ± 2.87	
Factor3, bulimia and food occupation					
Boys	0	0–2	6.133 *	1.35 ± 2.39	1.495
Girls	0	0–1		1.12 ± 2.23	
Total	0	0–2		1.24 ± 2.32	
Factor 4, social pressure to eat					
Boys	1	0–3	3.850 *	2.17 ± 2.78	−1.884
Girls	2	0–4		2.53 ± 2.95	
Total	1	0–4		2.33 ± 2.86	

Note: * *p* < 0.05; ** *p* < 0.01.

**Table 5 ijerph-20-00738-t005:** Correlations between the BMI and the ChEAT scores in boys and girls.

Factor	Total Sample		Girls		Boys	
	*r*	*p*	*r*	*p*	*r*	*p*
Factor 1 Fear of getting fat	0.268	<0.001	0.355	<0.001	0.309	<0.001
Factor 2 Dieting and purging behaviors	0.140	<0.001	0.160	0.002	−0.105	0.043
Factor 3 Bulimia and Food preoccupation	0.052	0.136	0.065	0.198	0.064	0.224
Factor 4 Social pressure to eat	−0.202	<0.001	−0.161	0.001	−0.233	<0.001
Total score	0.114	0.002	0.184	0.001	0.092	0.079

**Table 6 ijerph-20-00738-t006:** ChEAT scores by children’s BMI groups.

Factor	BMI Categories						Kruskal–Wallis	
ChEAT	Thinness Grade 3 (Mean ± SD)	Thinness Grade 2 (Mean ± SD)	Thinness Grade 1 (Mean ± SD)	Fit (Mean ± SD)	Overweight (Mean ± SD)	Obesity (Mean ± SD)	χ^2^	*p*
Factor 1 Fear of getting fat	* T: 1.83 ± 2.83 ** G: 1.69 ± 3.04 *** B: 2.20 ± 2.49	T: 2.26 ± 3.44 G: 2.33 ± 3.99 B: 2.08 ± 1.66	T: 2.59 ± 3.01 G: 2.49 ± 2.60 B: 2.74 ± 3.58	T: 4.07 ± 4.56 G: 5.03 ± 4.82 B: 3.17 ± 4.12	T: 5.65 ± 5.01 G: 6.81 ± 4.77 B: 4.86 ± 5.05	T: 5.68 ± 5.18 G: 5.76 ± 5.08 B: 5.63 ± 5.28	T: 45.412 G: 42.595 B: 20.511	T: <0.0001 G: <0.0001 B: 0.001
Factor 2 Dieting and purging behaviors	T: 0.63 ± 0.83 G: 0.43 ± 0.76 B: 1.20 ± 0.84	T: 0.67 ± 1.22 G: 0.54 ± 1.14 B: 0.93 ± 1.38	T: 1.08 ± 2.24 G: 0.67 ± 1.14 B: 1.71 ± 3.18	T: 1.28 ± 2.70 G: 1.26 ± 2.14 B: 1.29 ± 3.15	T: 1.73 ± 2.84 G: 1.89 ± 3.08 B: 1.62 ± 2.68	T: 2.15 ± 3.64 G: 2.29 ± 4.88 B: 2.08 ± 2.80	T: 16.085 G: 12.375 B: 11.672	T: 0.007 G: 0.030 B: 0.040
Factor 3 Bulimia and food occupation	T: 1.12 ± 2.37 G: 0.58 ± 1.73 B: 2.40 ± 3.36	T: 1.00 ± 1.49 G: 1.09 ± 1.60 B: 0.79 ± 1.25	T: 0.81 ± 1.64 G: 0.55 ± 1.24 B: 1.21 ± 2.06	T: 1.26 ± 2.53 G: 1.28 ± 2.62 B: 1.25 ± 2.45	T: 1.17 ± 2.02 G: 1.35 ± 2.16 B: 1.06 ± 1.92	T: 1.45 ± 2.33 G: 0.88 ± 1.64 B: 1.73 ± 2.56	T: 4.421 G: 6.770 B: 5.222	T: 0.491 G: 0.238 B: 0.389
Factor 4 Social pressure to eat	T: 3.26 ± 3.31 G: 2.86 ± 3.08 B: 4.40 ± 4.04	T: 3.56 ± 2.86 G: 3.26 ± 3.00 B: 4.33 ± 2.39	T: 2.90 ± 2.90 G: 3.10 ± 2.96 B: 2.63 ± 2.83	T: 2.36 ± 2.90 G: 2.42 ± 2.82 B: 2.31 ± 2.98	T: 1.88 ± 2.73 G: 2.25 ± 3.00 B: 1.63 ± 2.52	T: 1.83 ± 2.55 G: 2.29 ± 2.88 B: 1.58 ± 2.34	T: 28.778 G: 7.595 B: 22.319	T: <0.0001 G: 0.180 B: 0.001
Total score of ChEAT	T: 6.41 ± 6.23 G: 4.83 ± 5.94 B: 10.20 ± 5.72	T: 8.05 ± 6.62 G: 7.63 ± 7.54 B: 9.09 ± 3.59	T: 6.64 ± 6.24 G: 6.65 ± 4.16 B: 6.62 ± 8.78	T: 8.90 ± 9.74 G: 10.06 ± 9.37 B: 7.83 ± 9.98	T: 9.67 ± 8.87 G: 11.93 ± 9.40 B: 7.93 ± 8.08	T: 10.05 ± 8.68 G: 10.14± 9.35 B: 10.00± 8.39	T: 8.385 G: 13.357 B: 10.436	T: 0.136 G: 0.020 B: 0.064

Note: * T for total sample; ** G for girls; *** B for boys.

## Data Availability

Not applicable.

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
