# Peer review of "The Reliability and Validation of the Children’s Eating Attitude Test among Chinese Samples"

_ijerph, 2022, doi:10.3390/ijerph20010738_

Round 1

Reviewer 1 Report

This is a well-written, clear and interesting study, relating to the validation of the ChEAT tool on younger Chinese children. It adds to the data relating to measuring disordered eating amongst different groups and will be of interest to those working in this field. I have no reservation in recommending it for publication with only minor spelling and grammar checks needed, as there are some minor 'typos' in the manuscript e.g. line 102 change: problem (singular) to problems (plural). 

Author Response

Response: Thank you for the valuable suggestion. We have checked and revised the grammatical and spelling errors.

Reviewer 2 Report

I congratulate the authors on their choice of research topic and the Children's Eating Attitude Test (ChEAT) validation. The issues addressed in the article are very interesting and of interest to the reviewer. The study was carried out on a large sample of children and their parents.

In nutrition research, the relevance of a method means its ability to correctly identify an eating behaviour, nutrient or food intake, while the reliability of a method determines the accuracy of that identification. The higher the reliability, the greater the accuracy with which the method measures the trait and the lower the measurement error.

The purpose of this study was to examine the psychometric properties of the Children’s Eating Attitude Test (ChEAT) in a Chinese sample. The Chinese version of the ChEAT showed evidence for reliability and validity to evaluate the eating attitudes and behaviors for Chinese children. The eating behavior differed significantly among different genders and BMI groups.

In my opinion:

• A clear introduction is used in the manuscript;

• The material and research methods are described in detail;

• Research has been properly planned;

• Appropriate analytical methods were selected for the analysis of the results;

• The results of the analyzes are presented in graphical and tabular form.

I am asking for an answer to my doubts:

1.    The authors of the article said that in China, there was a lack of reliable and validated tools for screening eating disorders in children currently. Are other tests used in other countries (besides: Eating Attitudes Test - EAT, EDI - Eating Disorder Inventory and The Children’s Eating Attitude Test – ChEAT)?

2.    Why is regression analysing rarely used in validation studies?

Author Response

  1. The authors of the article said that in China, there was a lack of reliable and validated tools for screening eating disorders in children currently. Are other tests used in other countries (besides: Eating Attitudes Test - EAT, EDI - Eating Disorder Inventory and The Children’s Eating Attitude Test – ChEAT)?

Response: Thank you for the valuable questions. There are other tests used in other countries for screening eating disorders in children, such as the Child Eating Behavior Questionnaire (CEBQ). We added it to the section introduction:

For screening eating disorders in children, the Child Eating Behavior Questionnaire (CEBQ) is a parent-report questionnaire designed in the U.K. It included responsiveness to food, enjoyment of food, satiety responsiveness, slowness in eating. fussiness, emotional overeating, emotional undereating[19].

  1. Why is regression analysis rarely used in validation studies?

Response: Thank you for the valuable question. Regression analysis can only show how one variable changes with the others. Sometimes a good regression coefficient can be shown even if the two variables are very different in value. Therefore, it is appropriate to use agreement analyzing to assess the validation of questionnaires, scales and so on. We have addressed this limitation in the last paragraph discussion section as follows:

For concurrent validity, correlation analysis between the ChEAT and the BMI was not comprehensive for validity test, the future studies are encouraged to evaluate the concurrent validity of the Chinese version of the ChEAT by using multiple tools to assess eating disorders in children.

Reviewer 3 Report

Please see the attachment file. 

Author Response

  1. The manuscript examined the validity and reliability of the Children’s Eating Attitude Test in central urban or suburban areas of Shanghai, China. Also, the ChEAT difference between gender and BMI groups was assessed.
  2. Title “The Reliability and Validation of …………” suggest that the Validation can revised as Validity.

Response: Thank you for the valuable suggestion. We have revised our title as you suggested as follows: The Reliability and Validity of the Children’s Eating Attitude Test among Chinese children.

  1. Page1, Introduction part, line 4,“……to the DSM-5……”What is DSM-5? Please explain.

Response: Thank you for the valuable suggestion. It is an abbreviation of the Diagnostic and Statistical Manual of Mental Disorders, Fifth Edition. We have added it to the introduction section.

  1. P.2, third paragraph, The Children Eating Attitude Test (ChEAT) was a questionnaire

measure a wide range of problematic eating attitude and behaviors among children and

adolescents age 8-13 years. It not only measure attitude, but also behavior. However, the title:”……….and gender difference of eating behavior among Chinese children” Actually, the eating behavior was measure by ChEAT questionnaire. The audiences might confuse ChEAT measure only attitude not behavior. The eating behavior was tested by different questionnaire.

Response: Thank you for the valuable suggestion. As response for your second question., we had revised title.

  1. P.2 Second part, 2.1 participant. Line 4 mentioned “The choice of schools was random to preserve………..” Please explain more details about the procedure of random sample selection.

Response: Thank you. We have added the details as follows:

A total of 906 children (53.85% boys and 46.2% girls) from grades three to five and their parents were recruited by stratified cluster sampling…. The schools selected by stratified random sampling were located in central urban or suburb areas of Shanghai and represent different socioeconomic levels…. Informed consent was obtained from primary schools and legal parents of all participating students.

  1. P.7, Table 3, the Cronbach’s α off our dimension were from 0.536-0.728. However, the rule of thumb applies to most situation Cronbach’sαshould be >0.7 to be acceptable. The Cronbach’sα>0.5 is poor and >0.6 is questionable. F2, F3 and F4 was lower than 0.7, might need to be improved.

Response: Thank you for the consideration. We have addressed this limitation in the last paragraph discussion section as follows:

Our Cronbach's alpha values for Factors 2–4 were lower than 0.7 (0.536-0.699). A similar study conducted in Japan [28] showed similar results, Cronbach's alphas for Factors 2–5 between 0.39 and 0.62. The sample in this study comprises students with typical development, so the reliability for these factors are relatively low. We did not perform structured diagnostic for participants scored above cut-off to distinguish types of eating disorder, such as anorexia nervosa, bulimia nervosa, and binge eating disorder, which does not permit us to draw further conclusions about the scale’s validity in samples with specific eating disorder.

  1. P.9 Table 5, “Note:*p<0.0.5, **p<0.01”. The sings of * and ** should include in the table, such as “Factor1 r value is 0.268**(p<0.001)”
  2. P.10 Table6, The sings of * and ** should also include in the table.

Response: Thank you for the valuable suggestions. we had revised it.

  1. The contribution of this manuscripts should mention more. The practical implication, such as the results can assist which groups of people…..etc. and the academic implication should both include in this study.

ResponseThank you for the consideration. We have addressed contribution of this manuscripts in the conclusion section as follows:

The Chinese version of the ChEAT is a reliable and valid psychometric tool that may be useful in the assessment of children aged 9 to 13 years with abnormal eating attitudes. Our findings will facilitate cross-cultural comparative studies on eating disorders in children.